# TB Antigen-Stimulated CXCR3 Ligand Assay for Diagnosis of Tuberculous Lymphadenitis

**DOI:** 10.3390/ijerph18158020

**Published:** 2021-07-29

**Authors:** Wou-Young Chung, Keu-Sung Lee, Joo-Hun Park, Yun-Jung Jung, Seung-Soo Sheen, Ji-Eun Park, Joo-Sung Sun, Young-Hwa Ko, Kwang-Joo Park

**Affiliations:** 1Department of Pulmonology and Critical Care Medicine, Ajou University Hospital, Suwon 16499, Korea; biscut@ajou.ac.kr (W.-Y.C.); plator@aumc.ac.kr (K.-S.L.); lungmd@ajou.ac.kr (J.-H.P.); tomato81@aumc.ac.kr (Y.-J.J.); ssheen@ajou.ac.kr (S.-S.S.); petitprince012@ajou.ac.kr (J.-E.P.); 2Department of Radiology, Ajou University Hospital, Suwon 16499, Korea; sunnahn@ajou.ac.kr; 3Department of Pathology, Ajou University Hospital, Suwon 16499, Korea; rallo99@ajou.ac.kr

**Keywords:** tuberculosis, tuberculous lymphadenitis, CXCR3 ligand

## Abstract

The diagnosis of tuberculous lymphadenitis (TB-LAP) is challenging. We evaluated the role of blood CXC chemokine receptor 3 (CXCR3) ligands in its diagnosis. A total of 65 lymphadenopathy patients were enrolled and lymph node sampling was performed. We also recruited 113 control subjects, consisting of 27 with positive results and 86 with negative results, in the interferon (IFN)-γ release assay (IGRA). In all study subjects, whole-blood samples were collected using the IGRA methodology. After incubation, plasma levels of IFN-γ and two CXCR3 ligands, IFN-inducible T-cell a chemoattractant (I-TAC) and monokine induced by IFN-γ (MIG), were measured using immunoassay. Fifty-three TB-LAP patients were enrolled. TB antigen-stimulated IFN-γ, I-TAC, and MIG levels were all significantly higher in the TB-LAP patients than in the controls and non-TB-LAP patients. The levels of I-TAC and MIG, but not IFN-γ, showed significant differences between the TB-LAP patients and IGRA-positive controls. Area under the receiver operating characteristic curves (AUROCs) of IFN-γ, I-TAC, and MIG were 0.955, 0.958, and 0.959, respectively, for differentiating TB-LAP from control group, and were 0.912, 0.956, and 0.936, respectively, for differentiating TB-LAP from non-TB-LAP. In conclusion, the TB antigen-stimulated MIG and I-TAC could be useful biomarkers in the diagnosis of TB-LAP.

## 1. Introduction

Tuberculosis (TB) is still a major life-threatening disease throughout the world, and was responsible for 1.5 million deaths in 2010 [1]. Extrapulmonary TB (EPTB), the most common form of which is tuberculous lymphadenitis (TB-LAP), accounts for 15–20% of TB cases [2]. The diagnosis of EPTB is challenging because of its nonspecific clinical features and the difficulty to demonstrate the presence of microorganisms in affected tissues. Fine-needle aspiration (FNA) is a simple and cost-effective method for this purpose. However, smear microscopy of FNA samples has low sensitivity in lymph node specimens due to the paucibacillary nature of the disease [3], and even a positive smear result has limited specificity with the emergence of nontuberculous mycobacteria (NTM) [4,5]. Tissue culture for *Mycobacterium tuberculosis* is highly specific but has low sensitivity and requires several weeks to obtain a result. Histological examination of FNA samples is a key method for diagnosis of TB involvement of the tissue, but it is difficult to distinguish from other granuloma-forming conditions, such as NTM infection, sarcoidosis, Kikuchi’s disease, nocardiasis, and a variety of foreign body reactions [6]. Molecular technologies, such as polymerase chain reaction (PCR), can significantly improve diagnostic yield [7]. More invasive procedures, such as excisional biopsy, which has good specificity, can yield higher sensitivity but still only reach between 60% and 80% [8]. Moreover, the high prevalence of EPTB in developing countries with limited medical resources makes invasive diagnostic procedures less feasible. Therefore, a simple, noninvasive, accurate diagnostic tool is required. Blood tests are safe and the results can be available within hours to a few days. The interferon (IFN)-γ release assay (IGRA) is the major blood test for detection of TB. IGRA has overcome the cross-reactivity with NTM, which was a major drawback of the tuberculin skin test, and it has been used extensively for over a decade. The diagnostic performance of IGRA in TB-LAP has also been widely studied [9,10,11,12]. The reported sensitivity and specificity of IGRA for the diagnosis of TB-LAP range from 80% to 92% and 68% to 95%, respectively. However, IGRA still has limitations as it cannot distinguish active disease from latent infection [13,14], and in South Korea, about one third of the population is latently infected with TB [15,16]. Ligands for CXC chemokine receptor 3 (CXCR3), which act downstream of IFN-γ and regulate Th1 pathways [17], have been evaluated for their applicability in the diagnosis of TB [18,19]. CXCR3 ligands consist of monokine induced by IFN-γ (MIG, CXCL9), IFN-γ-inducible 10 kDa protein (IP-10, CXCL10), and IFN-inducible T-cell α-chemoattractant (I-TAC, CXCL11). IP-10 is the most widely evaluated but does not show superiority to IFN-γ for diagnosis of active TB [20,21] and fails to distinguish active TB from latent disease [22]. We previously found that I-TAC and MIG levels in blood may be useful markers for diagnosing [18] and monitoring the treatment response in active pulmonary TB [23,24]. Furthermore, in tuberculous pleurisy, which is also a common form of extrapulmonary TB, CXCR3 ligands in pleural fluid can be used as surrogate markers of the disease [25].

Here, we evaluated the roles of TB antigen-stimulated levels of I-TAC and MIG in the diagnosis and clinical assessment of TB-LAP in comparison with IFN-γ.

## 2. Materials and Methods

Between August 2012 and July 2018, we recruited adult patients with suspected TB-LAP at Ajou University Hospital, a referral tertiary hospital. Sixty-five patients agreed to participate in the study. The patients had cervical (*n* = 58), intrathoracic (*n* = 5), and intraabdominal (*n* = 2) involvement. Controls consisting of 113 healthy individuals without known active disease or recent contact with active pulmonary TB were recruited from among those undergoing routine checkups at the Ajou University Health Examination Center between January 2017 and December 2018. The control group was divided into IGRA-positive (*n* = 27) and IGRA-negative (*n* = 86) subgroups.

The study was approved by the Ajou University Hospital Institutional Review Board (approval number MED-SMP-12-068). All subjects provided written informed consent.

### 2.1. Lymph Node Sample Collection

Lymph node sampling was performed in all patients. For 58 patients with cervical lymphadenitis, fine-needle aspiration (*n* = 6), core-needle biopsy (*n* = 45), and excisional biopsy (*n* = 7) were performed. Mediastinoscopy with biopsy was performed in five patients with intrathoracic lymphadenitis. Peritoneoscopic biopsy was performed in two patients with intraabdominal lymphadenitis.

### 2.2. Mycobacterial Culture

Biopsy samples were homogenized in a sterile mortar and pestle and digested and decontaminated using the N-acetyl-l-cysteine (NALC)-NaOH method. Ziehl–Neelsen staining was performed on tissue smears directly after homogenization and after concentration. Specimens were cultured for *M. tuberculosis* on liquid modified Middlebrook 7H9 medium (BacT/ALERT MP System; bioMérieux, Durham, NC, USA) and/or on 3% solid Ogawa medium (Eiken, Tokyo, Japan).

### 2.3. Histopathology

Formalin-fixed lymph node biopsy tissues were processed on an automated tissue processor using a 12 h cycle with 10% formalin, isopropyl alcohol, xylene, and paraffin wax. Paraffin blocks of the processed tissue were cut into sections 3–5 mm thick, which were stained with hematoxylin and eosin for histopathological analysis.

### 2.4. Polymerase Chain Reaction

PCR was performed on fine-needle aspiration samples or deparaffinized tissue samples of lymph nodes. All PCR steps were performed in separate rooms to avoid carryover of templates. After extraction of DNA, two-step nested multiplex PCR was performed using commercially available PCR kit (GeNei^TM^, Bangalore Genei, Bangalore, India). For identification and differentiation of TB and NTM, we used primer pairs to amplify *IS6110*, *MTP40*, and *32kD α-antigen* gene sequences for *M. tuberculosis* complex, *M. tuberculosis*, and NTM, respectively (Table 1). *IS6110* is commonly used target sequence for TB. However, *IS6110* is not a specific gene for *M. tuberculosis*, because it is present in all the strains of the *M. tuberculosis* complex, which includes the human pathogenic species *M. tuberculosis*, *M. bovis*, and *M. africanum*. Furthermore, it was reported that some strains of *M. tuberculosis* do not possess the *IS6110* target sequence in their genome, which may cause false negative results. Therefore, additional identification of *M. tuberculosis*-specific *MTP40* can play a complementary role to increase diagnostic accuracy. *32kD α-antigen* is present in all mycobacterial species, so it can be a useful marker for NTM when considered along with the PCR results of the above two genes for *M. tuberculosis* complex and *M. tuberculosis* [26].

### 2.5. Evaluation and Treatment of TB

Patients with TB-LAP were divided into three groups according to the criteria of the European Centre for Disease Prevention and Control (ECDC): confirmed (culture-positive or culture-negative but microscopy- and PCR-positive), probable (culture-negative but microscopy- or PCR-positive or pathology suggestive of TB), and possible (based solely on clinical assessment) [27]. NTM lymphadenitis was diagnosed in cases where NTM was identified in specimen culture or microscopy findings of the tissue pathology compatible with mycobacterial infection were accompanied by positive NTM PCR and negative TB PCR [28]. As computed tomography (CT) of the affected area was performed in all patients, we analyzed the number and sites of involved lymph nodes. CT criteria for lymphadenopathy was defined as one of the following criteria was fulfilled: (a) size, short diameter was larger than 1 cm (1.5 cm for neck level II); (b) shape, loss of normal coffee bean shape; or (c) presence of necrosis, calcification, or perinodal infiltration [29]. Cervical lymph node levels were classified according to currently widely accepted classification: Level I, submental (IA) and submandibular (IB); level II, upper internal jugular nodes; level III, middle jugular nodes; level IV, low jugular nodes; level V, posterior triangle nodes; level VI, upper visceral nodes; level VII, superior mediastinal nodes [30]. A diagnosis of concomitant pulmonary TB was made if *M. tuberculosis* was identified in sputum culture or appropriate chest radiological features were accompanied by appropriate treatment responses.

The cases were reviewed independently by two pulmonary medicine physicians, one radiology specialist and one pathologist.

The antituberculous medication choice and management of the patients were based on the Korean Guidelines for TB 2011 [31]. Treatment was initiated using a standard regimen of rifampicin, isoniazid, pyrazinamide, and ethambutol. The minimum duration of therapy was 6 months. All patients were followed up monthly by clinical examination, laboratory blood tests, chest radiography, and sputum examination. The regimen and duration of therapy were modified at the discretion of the physician in charge according to the treatment response, patient compliance, and presence of adverse events or drug resistance.

### 2.6. Plasma Sampling and Measurement of Marker Levels

Blood samples were obtained from suspected TB patients before commencement of antituberculous medication. Samples of 1 mL blood were drawn and placed into three tubes supplied with the QuantiFERON-TB GOLD In-Tube test kit (Qiagen, Dusseldorf, Germany). The tubes were precoated with saline (nil tube) or ESAT-6, CFP10, and TB 7.7 (TB-antigen tube) or phytohemagglutinin (mitogen tube). The tubes were incubated for 18 h at 37 °C, and plasma was harvested and frozen until further analysis. I-TAC and MIG were measured in the samples collected using a Quantikine enzyme-linked immunosorbent assay (ELISA) kit (R&D Systems, Minneapolis, MN, USA). IFN-γ levels were measured using the QuantiFERON-TB Gold In-Tube (Qiagen) ELISA.

### 2.7. Statistical Analysis

Data analyses were performed using R (version 3.1.2; R Foundation for Statistical Computing, Vienna, Austria). Data are presented as the median (interquartile range) due to the non-normal distribution. The Kruskal–Wallis test was used for comparisons among three or more groups followed by Dunn’s post hoc analysis for pairwise multiple comparisons.

To determine the diagnostic accuracy of the markers, receiver operating characteristic (ROC) analysis was performed, and area under the curve (AUC) was calculated. AUC represents the probability that the test can discriminate between subjects with and without TB-LAP. AUC was calculated by the trapezoidal rule, based on the Mann–Whitney two-sample statistic (asymptotic theory developed for U-statistics). The AUC values range from 0 to 1, with an AUC of 1 indicating 100% probability that a given test can discriminate the disease status. Sensitivity and specificity were calculated using the usual equations: sensitivity = number of true positives/(number of true positives + number of false negatives); specificity = number of true negatives/(number of true negatives + number of false positives). Optimal cutoff value was determined using Youden index, which combines sensitivity and specificity by evenly maximizing them into a single measure (Sensitivity + Specificity − 1) [32,33].

## 3. Results

### 3.1. Participant Characteristics

A total of 65 patients with suspected TB-LAP were recruited into the study. Forty-seven patients with cervical lymphadenitis were referred to the Pulmonology Clinic from the Head and Neck Division of the Ear-Nose Throat Department of Ajou University Hospital. Eleven others were referred from community clinics. Three thoracic lymphadenitis patients were referred from the Thoracic Surgery Department and two others were referred from the Health Promotion Department. Two abdominal lymphadenitis patients came to the Pulmonology Clinic from the Hepatology Division of the Gastroenterology Department. A diagnosis of active TB-LAP was finally made in 53 patients, all of whom received antituberculous treatment. The demographic and clinical characteristics of participants are shown in Table 2.

Among 58 cervical lymphadenitis patients, 47 were diagnosed with TB-LAP. Nontuberculous causes included NTM (*n* = 6), self-limited inflammatory lymphadenitis (*n* = 1), and metastatic malignancy (*n* = 4). Among five patients with thoracic lymphadenitis, one was diagnosed with TB-LAP. Other nontuberculous causes included sarcoidosis and granulomatous mastitis in two cases each. Two patients with intraabdominal lymphadenitis were found to have TB-LAP. Representative CT images of TB-LAP are presented in Figure 1.

Among 53 TB-LAP patients, the number of involved lymph nodes was ≥4 in 31 patients. The number of involved levels was ≥3 in 26 patients. Thirteen patients had combined pulmonary TB (Table 3).

We also recruited 113 control subjects, 86 of whom were IGRA-negative and 27 were IGRA-positive. The 27 IGRA-positive controls were classified as having latent TB. Forty-eight patients showed improvement of symptoms and radiological remission after six months of standard TB treatment. Five others (cervical TB-LAP, *n* = 3; thoracic TB-LAP, *n* = 1; and abdominal TB-LAP, *n* = 1) achieved remission at nine months.

### 3.2. Comparison of CXCR3 Ligands and IFN-γ Levels

TB antigen-stimulated IFN-γ, I-TAC, and MIG levels were significantly higher in the TB-LAP group than in the control and non-TB-LAP groups. The levels of I-TAC and MIG, but not IFN-γ, were significantly different between the TB-LAP group and the IGRA-positive control group (Figure 2). Similar results were observed after selecting only the group with a definite diagnosis of TB (Figure 3). By detailed pairwise comparisons of the groups, I-TAC levels showed more significant differences between TB-LAP patients and other groups than MIG levels did. With regard to differentiating TB-LAP patients from LTBI, I-TAC showed significant results in both all and definite patient groups, while IFN-γ showed only a trend toward significant difference in all TB-LAP patients (*p* = 0.099), and MIG showed only a trend toward significant difference (*p* = 0.010) in definite TB-LAP patients.

Exact *p*-values of Dunn’s pairwise comparisons from the Kruskal–Wallis tests can be informative and presented in Appendix A. 

The marker levels were slightly higher in TB-LAP patients with than without pulmonary TB, but the differences were not significant. The anatomical extent of lymph node involvement, such as number of lymph nodes or multiple-level invasion, was not associated with significant changes in marker levels, although MIG tended to be elevated in patients with enlargement of more than four lymph nodes (Table 3).

### 3.3. ROC Analysis

TB-antigen stimulated levels of I-TAC and MIG showed better performance than IFN-γ for diagnosis of TB-LAP. The AUCs of I-TAC and MIG were greater than those of IFN-γ in all situations (Table 4; Figure 4). The AUC, specificity, and sensitivity of CXCR3 ligands were better after selecting only the group with a definite diagnosis of TB, and I-TAC showed the best diagnostic capability (Table 5; Figure 5).

### 3.4. Co-Application of the Markers by Combination

The combination of the markers may improve predictive performance over individual markers. When we divided the TB-LAP patients using the cutoff values of the markers for differentiating the TB-LAP patients from the controls, six patients had low levels of I-TAC, and five patients of them showed high levels of MIG. Likewise, I-TAC levels were high in three of the four patients with low levels of MIG. 

Furthermore, we evaluated the cohort of high IFN-γ expressing controls who are IGRA positive, using the cutoff values of the markers for differentiating the TB-LAP patients from the IGRA^+^ controls. As a result, among 28 subjects with IGRA^+^ controls, 20 subjects had low MIG levels and 22 subjects had low I-TAC levels. These results show potential utility of the markers to distinguish active TB patients from LTBI subjects, although the results may also vary according to the applied cutoff values.

## 4. Discussion

The TB antigen-stimulated whole blood assay using MIG and I-TAC was useful for diagnosing TB-LAP compared to non-TB-LAP as well as controls. The overall performance of the TB antigen-stimulated CXCR3 ligand assay was superior to IGRA, and CXCL11 showed the highest discriminating power. In addition, it was promising that the assay showed good capability to distinguish TB-LAP patients from IGRA-positive controls. Our results indicate slightly lower but comparable diagnostic performance of TB-antigen-stimulated CXCR3 ligands levels to the previous studies on pulmonary TB [18,19,23], and showed similar performance to a study on EPTB tuberculous pleural effusion [25].

By the combination of the markers, we found the mutually complementary roles of the studied markers to enhance diagnostic performance for TB-LAP, along with aforementioned discriminating power of the CXCR3 ligands between TB-LAP and IGRA-positive controls. In this regard, co-application of IGRA and CXCR3 ligands using a panel can be a promising tool for the diagnosis of tuberculosis, especially for differentiation between active tuberculosis and LTBI. The validity of their co-application can be more significant in countries with high prevalence of LTBI. As a prerequisite for better use of the studied markers along with IFN-γ, the cutoff values should be established through further case-control studies.

Theoretically, the circulating levels of CXCR3 ligands, as Th1 immunity-related mediators, may be lower in EPTB than in pulmonary TB. The level of the T cell immune response is correlated with the number of bacteria in affected tissue [34], and the viable bacillary load is much higher in pulmonary TB than TB-LAP. Furthermore, organized well-formed granulomas found in TB-LAP indicate effective immune control against *M. tuberculosis*, and the levels of secreted TB antigens have been reported to be lower in TB-LAP than in pulmonary TB [35]. However, in our study, the TB antigen-stimulated CXCR3 levels in TB-LAP patients without pulmonary TB were slightly lower but not significantly different from those in patients with pulmonary TB. These observations may be attributed to the generally low severity of pulmonary TB cases and small sample size in this study. Further studies on larger populations are required for clarification.

Tissue expression and blood levels of CXCR3 ligands have been evaluated in EPTB patients. One study showed that the expression levels of CXCR, IFN-γ, MIG, and IP-10 were increased and their blood concentrations were significantly higher in 36 patients with spinal TB than in healthy controls [36]. In addition, immunohistochemical analyses have demonstrated overexpression of MIG, IP-10, and I-TAC especially in granulomas in 26 patients with tubal TB, and their expression levels are correlated with disease severity [37]. Another study reported that unstimulated blood levels of IP-10 are correlated with paradoxical reaction in TB-LAP [38]. TB antigen ESAT6-induced MIG release is increased in TB-LAP and correlated with severity of disease [39].

IGRA is recognized as the most potent biomarker for diagnosis of TB, showing superior sensitivity and specificity to the tuberculin skin test [14,40,41]. For EPTB diagnosis, IGRA show variable sensitivity ranging from as low as 34% to 95% [42,43,44], and its utility for the diagnosis of active disease is reduced in highly endemic areas because of the high prevalence of LTBI. There is a paucity of data for TB-LAP, which is the most common form of EPTB [12,44]. The diagnostic performance of IGRA in EPTB is limited in Korea where TB has an intermediate prevalence and one third of adults are latently infected. The relatively low capability of IGRA to discriminate between active and latent disease and high prevalence of LTBI in Korea reduce the reliability of IGRA as a biomarker for EPTB, especially in adult patients.

In this study, I-TAC and MIG levels were significantly higher in the TB-LAP group than the non-TB-LAP group as well as both IGRA-positive and IGRA-negative controls. IFN-γ was significantly higher in active TB-LAP compared to IGRA-negative control and non-TB-LAP groups but did not show a significant difference compared to IGRA-positive controls, which can make it difficult to use in regions of high TB prevalence where the rate of latent TB infection is high.

The utility of CXCR3 ligands in active TB diagnosis and the superior diagnostic value compared to IFN-γ can be explained as follows. First, in the immune reaction cascade initiated by *M. tuberculosis*, IFN-γ stimulates various cell populations, including natural killer cells, macrophages, and neutrophils, and activates the Th1 immune response. The CXCR3 ligands induced by IFN-γ more specifically activate Th1 responses and antagonize the Th2 response by acting on CCR3, and the actions of CXCR3 ligands are more downstream than those of IFN-γ. Second, the concentrations of I-TAC and MIG were 10–20-fold higher than the concentration of IFN-γ. These differences could also enhance their utility as clinical biomarkers.

In this study, the CXCR3 ligands and IFN-γ levels were poorly correlated with the anatomical extent of disease. They were slightly increased in TB-LAP with involvement of multiple levels and invasion of large numbers of lymph nodes, but the differences were not significant. These poor correlations with severity or extent of disease should be reassessed in further studies with larger populations. However, gross anatomical extent of disease does not necessarily reflect the degree of ongoing inflammation.

This study had several limitations. Under real-world conditions, it is necessary to differentiate TB-LAP patients from non-TB-LAP patients more often than from healthy subjects. However, the number of patients with non-TB-LAP is small because patients with lymphadenitis were mostly referred from other departments, such as otolaryngology, after making a preliminary diagnosis of TB. In addition, the enrollment of suspected TB patients was made after application of considerable diagnostic procedures, which may surpass the clinical stage of “TB suspicion.” To better establish the diagnostic utility of the markers, the levels in TB patients should be compared with more non-TB-LAP cases in future studies.

There was a possibility of misclassification with regard to probable TB cases, despite the effort made for their correct classification. Although we attempted to correctly differentiate TB-LAP from non-TB-LAP cases, there may have been a chance of misclassification. However, definite TB-LAP cases also showed similar results. In addition, we rechecked the accuracy of clinico-radiological and pathological diagnoses taking into account the clinical course of all patients over time at the end of the study, and our initial categorization was shown to be correct.

The proportion of patients with LTBI was uncertain. IGRA positivity alone cannot define latent infection, and the investigation of TB contact may make a contribution but could not be incorporated because of the limited accuracy and vagueness of remote contact history. In addition, the tuberculin skin test was not included because of its limited usefulness due to routine Bacillus Calmette–Guérin (BCG) vaccination in Korea.

CXCR3 ligands can be triggered by a variety of stimuli. Therefore, it is difficult to make a diagnosis of TB based solely on blood levels of chemokines. However, they could be used as simple, cheap, and reliable adjunctive clinical markers of mycobacterial infection.

## 5. Conclusions

The TB antigen-stimulated CXCR3 ligands MIG and I-TAC are significantly elevated in active TB-LAP with specificity and sensitivity superior to IGRA. Therefore, they could be useful as biomarkers in diagnosis of TB-LAP.

## Figures and Tables

**Figure 1 ijerph-18-08020-f001:**
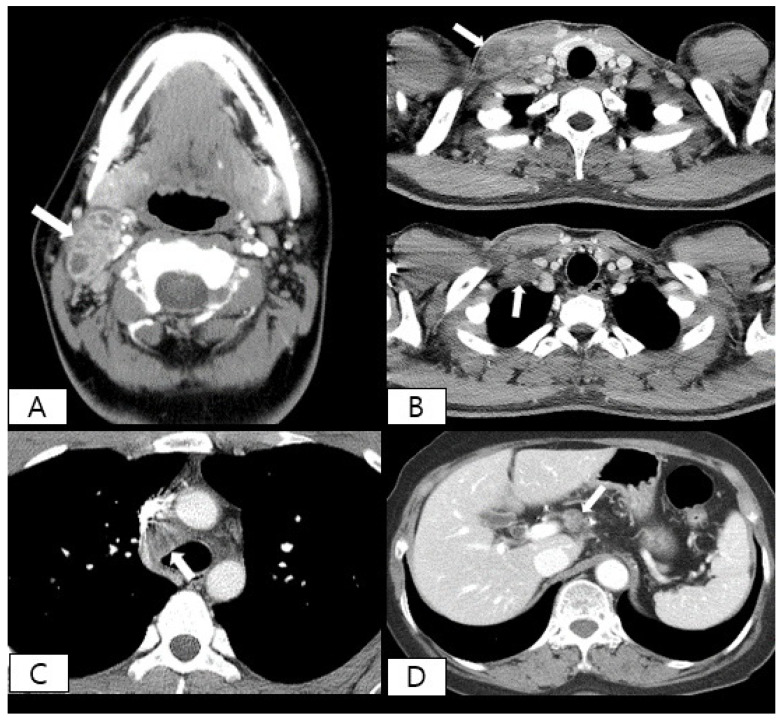
Post-contrast computed tomography (CT) images of four representative cases of pathologically proven tuberculous lymphadenitis. (**A**) Neck CT scan of 19-year-old female patient shows multiple conglomerated lymph nodes in level II on the right side of neck. Enlarged lymph nodes show peripheral rim enhancement and central necrosis (arrow). (**B**) Neck CT scan of 23-year-old male patient shows enlarged lymph nodes with necrosis (arrows) in level IV on the right side of neck (upper panel) and supraclavicular space (lower panel). (**C**) Chest CT scan of 31-year-old female patient shows enlarged lower paratracheal lymph nodes on the right (arrow). (**D**) Abdominal CT scan of 72-year-old female patient shows multiple enlarged lymph nodes in gastrohepatic ligament area (arrow).

**Figure 2 ijerph-18-08020-f002:**
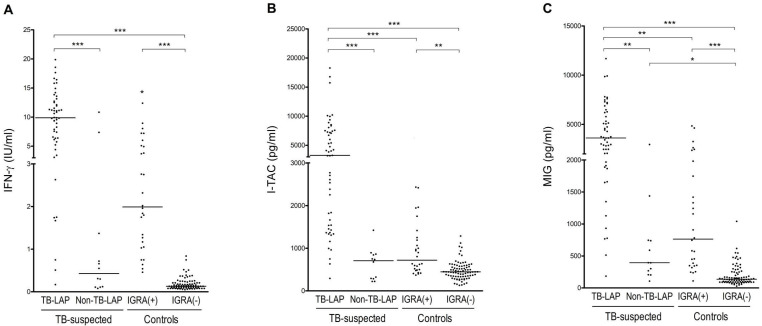
Comparison of tuberculosis-antigen-stimulated (**A**) interferon (IFN)-γ, (**B**) IFN-inducible T cell α chemoattractant (I-TAC), and (**C**) monokine induced by IFN-γ MIG) levels among all tuberculous lymphadenopathy (TB-LAP) patients (*n* = 53), non-tuberculous lymphadenopathy (Non-TB-LAP) patients, and IFN-γ release assay (IGRA)-positive and -negative control groups. Horizontal lines represent median values. * *p* < 0.05; ** *p* < 0.01; *** *p* < 0.001 (Kruskal–Wallis analysis with Dunn’s comparison test).

**Figure 3 ijerph-18-08020-f003:**
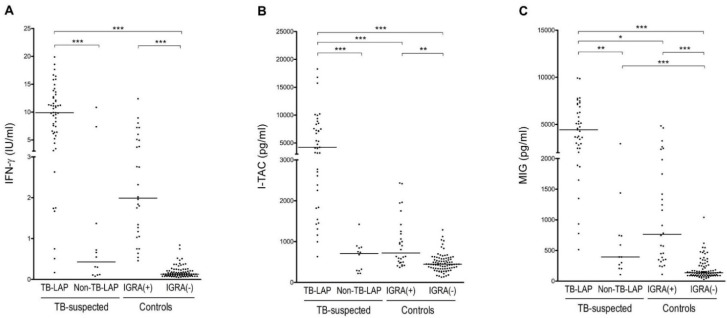
Comparison of tuberculosis-antigen-stimulated (**A**) interferon (IFN)-γ, (**B**) IFN-inducible T cell α chemoattractant (I-TAC), and (**C**) monokine induced by IFN-γ (MIG) levels among definite tuberculous lymphadenopathy (TB-LAP) patients (*n* = 39), non-tuberculous lymphadenopathy (Non-TB-LAP) patients, and IFN-γ release assay (IGRA)-positive and -negative control groups. Horizontal lines represent median values. * *p* < 0.05; ** *p* < 0.01; *** *p* < 0.001 (Kruskal–Wallis analysis with Dunn’s comparison test).

**Figure 4 ijerph-18-08020-f004:**
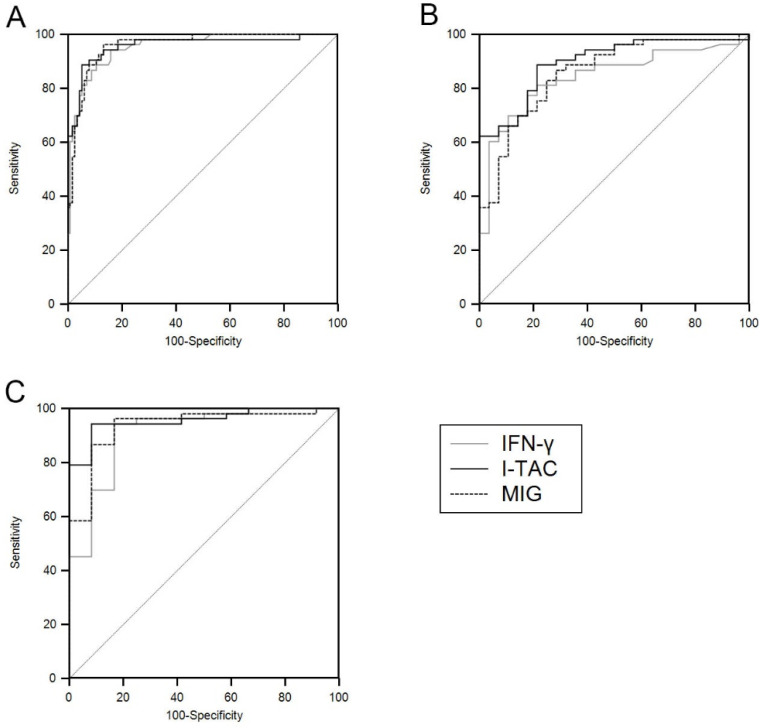
Receiver operating characteristic curves of interferon (IFN)-γ, IFN-inducible T-cell α–chemoattractant (I-TAC), and monokine induced by IFN-γ (MIG) for differentiating all tuberculous lymphadenitis patients from all controls (**A**), IFN-γ release assay-positive controls (**B**), and patients with non-tuberculous lymphadenitis (**C**).

**Figure 5 ijerph-18-08020-f005:**
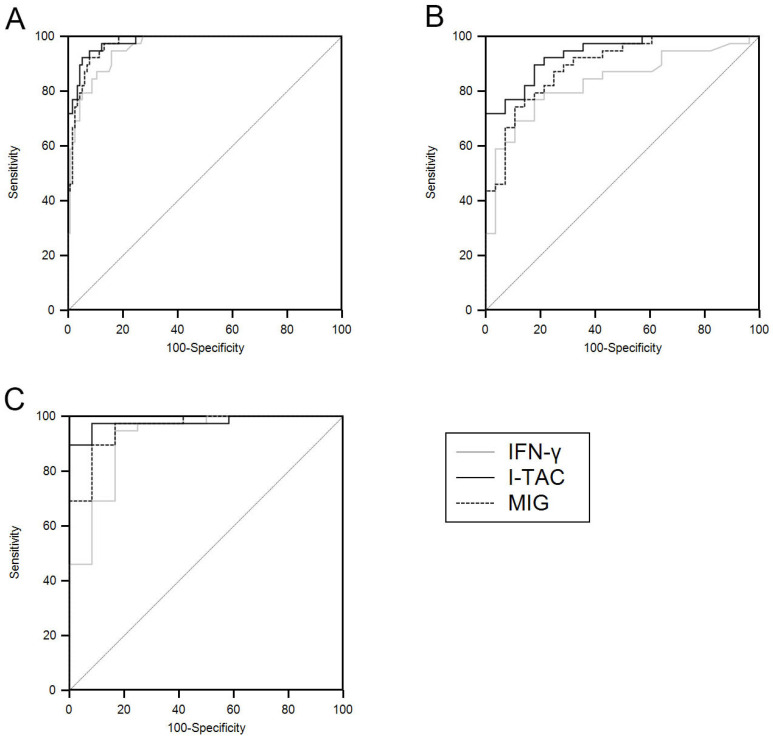
Receiver operating characteristic curves of interferon (IFN)-γ, IFN-inducible T-cell α–chemoattractant (I-TAC), and monokine induced by IFN-γ (MIG) for differentiating definite tuberculous lymphadenitis patients from the controls (**A**), IFN-γ release assay-positive controls (**B**), and patients with non-tuberculous lymphadenitis (**C**).

**Table 1 ijerph-18-08020-t001:** Sequences of primers used in nested multiplex PCR for the detection of *Mycobacterium* species.

Genes	Cycles	Primers	Sequence (5’-3’)	Size (bp)
*IS6110*	First round	MTBC F	CGGAGACGGTGCGTAAGTGG	984
		MTBC R	GATGGACCGCCAGGGCTTGC	
	Nested cycles	MTBC2F	CGATCGCCCCATCGACCTACT	500
		MTBC2 R	GGTCGAGTACGCCTTCTTGT	
*MTP40*	First round	MTB F	CGGCAACGCGCCGTCGGTGG	396
		MTB R	CCCCCCACGGCACCGCCGGG	
	Nested cycles	MTB2 F	CGTTCGGGATGCACTGCG	342
		MTB2 R	CACCCGGCGAATTCGTCAC	
*32kD* α*-antigen*	First round	NTM F	TTCCTGACCAGCGAGCTGCCG	506
		NTM R	CCCCAGTACTCCCAGCTGTGC	
	Nested cycles	NTM2 F	CACCCGCAGTTCATCTA	413
		NTM2 R	CGTTGTAGGCGTCCTGG	

**Table 2 ijerph-18-08020-t002:** Characteristics of the study population.

	TB-LAP	Non-TB-LAP	Controls	
			IGRA (+)	IGRA (–)
Subjects	53	12	27	86
Age (years)	41 (28–58)	45 (36–59)	42 (38–48)	40 (34–47)
Male	21 (39%)	5 (41%)	21 (77%)	49 (57%)
Smoking history	14(26%)	3 (25%)	9 (33%)	19 (21%)
Alcohol abuse history	4	2	0	0
Recent close contact with active TB	7	0	0	0
IGRA results				
Positive	46	5	27	0
Negative	4	6	0	86
Indeterminate	1			
Steroid use	3	2	0	0
Underlying conditions				
Diabetes mellitus	3	0	2	0
Malignancy	5	1	0	0
Immunosuppression	1	1	0	0
History of old TB	6	1	2	0

Data are medians (interquartile range) or *n* (%). TB-LAP = tuberculous lymphadenopathy; non-TB-LAP = non-tuberculous lymphadenopathy; IGRA = interferon- γ release assay; TB = tuberculosis.

**Table 3 ijerph-18-08020-t003:** TB antigen stimulated marker levels according to clinico-radiological features of TB-LAP patients.

***Pulmonary Tuberculosis* * **					
	Absent (*n* = 40)		Present (*n* = 13)		p-value
IFN-γ (IU/mL)	9.68	(5.91–12.21)	10.62	(7.41–15.81)	0.320
I-TAC (pg/mL)	3226.3	(1397.3–7298.9)	4751.3	(2254.0–8349.3)	0.272
MIG (pg/mL)	3313.2	(1900.5–6036.7)	4435.4	(3012.2–6203.5)	0.263
***Number of involved lymph node levels* ** **					
	<3 (*n* = 27)		≥3 (*n* = 26)		
IFN-γ (IU/mL)	9.78	(5.85–11.54)	9.88	(7.73–13.90)	0.335
I-TAC (pg/mL)	3518.2	(1543.1–8042.1)	3260.3	(1482.0–7343.9)	0.923
MIG (pg/mL)	3222.3	(1968.3–6203.5)	4285.3	(2268.9–6036.7)	0.678
***Number of involved lymph nodes* ^#^**					
	<4 (*n* = 22)		≥4 (*n* = 31)		
IFN-γ (IU/mL)	10.01	(5.90–11.97)	10.62	(6.98–12.83)	0.795
I-TAC (pg/mL)	3276.5	(453.5–7179.1)	3845.3	(1539.9–7463.0)	0.585
MIG (pg/mL)	2986.4	(1968.3–6203.5)	3752.4.	(2268.9–5910.1)	0.536

Data are medians (interquartile range). TB = tuberculosis; TB-LAP = tuberculous lymphadenitis; IFN = interferon; I-TAC = interferon-inducible T cell α chemoattractant; MIG = monokine induced by interferon-γ. * refer to the Material and Method section for the diagnostic criteria of concomitant pulmonary TB. ** refer to the Material and Method section for the classification of cervical lymph node levels. ^#^ refer to the Material and Method section for the CT criteria of lymphadenopathy.

**Table 4 ijerph-18-08020-t004:** Diagnostic performance of markers for differentiation of TB-LAP patients (*n* = 53) from those in other groups.

	AUC	95%CI	Cutoff	Sensitivity	Specificity
TB-LAP TB vs. all Controls					
IFN-γ	0.955	0.911–0.981	1.34 IU/mL	94.3	84.1
I-TAC	0.958	0.915–0.983	1287.2 pg/mL	88.7	94.7
MIG	0.959	0.917–0.984	745.7 pg/mL	96.2	86.7
TB-LAP vs. IGRA-positive Controls					
IFN-γ	0.841	0.743–0.913	5.09 IU/mL	81.1	78.6
I-TAC	0.896	0.808–0.953	1250.4 pg/mL	88.7	78.6
MIG	0.858	0.762–0.925	1419.6 pg/mL	86.8	71.4
TB-LAP vs. Non-TB-LAP					
IFN-γ	0.912	0.815–0.968	1.37 IU/mL	94.3	83.3
I-TAC	0.956	0.874–0.991	894.8 pg/mL	94.3	91.7
MIG	0.936	0.846–0.981	745.6 pg/mL	96.2	83.3

TB-LAP = tuberculous lymphadenitis; AUC = area under the curve; CI = confidence interval; IFN-γ =interferon-γ; I-TAC = interferon-inducible T cell α chemoattractant; MIG = monokine induced by interferon-γ; IGRA = interferon-γ release assay; non-TB-LAP = non-tuberculous lymphadenitis.

**Table 5 ijerph-18-08020-t005:** Diagnostic performance of markers for differentiation of definite TB-LAP patients (*n* = 39) from those in other groups.

	AUC	95%CI	Cutoff	Sensitivity	Specificity
TB-LAP vs. all Controls					
IFN-γ	0.958	0.913–0.984	1.34 IU/mL	94.9	84.1
I-TAC	0.981	0.944–0.996	1287.2 pg/mL	92.3	94.7
MIG	0.973	0.932–0.992	1330.7 pg/mL	92.3	92.0
TB-LAP vs. IGRA-positive Controls					
IFN-γ	0.835	0.724–0.914	6.01 IU/mL	76.9	82.1
I-TAC	0.939	0.852–0.982	1250.4 pg/mL	89.7	82.1
MIG	0.895	0.795–0.956	2634.4 pg/mL	74.4	89.3
TB-LAP vs. Non-TB- LAP					
IFN-γ	0.919	0.807–0.976	1.37 IU/mL	94.9	83.3
I-TAC	0.979	0.892–0.997	1422.1 pg/mL	89.7	100
MIG	0.959	0.863–0.994	1438.6 pg/mL	89.7	91.7

TB-LAP = tuberculous lymphadenitis; AUC = area under the curve; CI = confidence interval; IFN-γ = interferon-γ; I-TAC = interferon-inducible T cell α chemoattractant; MIG = monokine induced by interferon-γ; IGRA = interferon-γ release assay; non-TB-LAP = non-tuberculous lymphadenitis.

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
