# Peer review of "TB Antigen-Stimulated CXCR3 Ligand Assay for Diagnosis of Tuberculous Lymphadenitis"

_ijerph, 2021, doi:10.3390/ijerph18158020_

Round 1

Reviewer 1 Report

The manuscript entitled “TB antigen-stimulated CXCR3 ligand assay for diagnosis of Tuberculous Lymphadenitis” is a brief study into the use of circulating protein biomarkers in the differential diagnosis of extrapulmonary TB.  The authors demonstrate that CXCL11 and CXCL9 are significantly elevated in patients with TB-LAP as compared to non-TB-LAP patients.  Additionally they demonstrate a mild increase, as compared to IFN-g, for the diagnostic potential of these cytokines in differentiation of active vs latent TB, defined by IGRA positivity within the control arm.  Finally, the authors are complemented for clearly describing the limitations of their work and not presenting any overinterpretation of the data. Overall, this study adds to a growing body of biomarker discovery in the differential diagnosis of TB.   A few things that should be addressed prior to publication:

  • Please add primer sequences used for the PCR differentiation of TB and NTM
  • Discuss the possibility of using all three biomarkers as a panel. Do TB-LAP patients with low levels of circulating CXCL11 also demonstrate low levels of CXCL9?  Within the cohort of high IFN-g expressing controls who are IGRA+, is there high levels of CXCL9 and CXCL11?
  • Please Comment on the co-application of IGRA with plasma biomarker evaluation.

Author Response

We truly appreciate all the constructive comments and invaluable suggestions from the editor and the reviewers.

We tried to respond point by point to the remarks addressed by the reviewer.

The full responses to the remarks is in the attached file.

Best regards.

Reviewer 2 Report

In this paper Chung et. al. have proposed to use blood based CXCR3 ligands  for the diagnosis of tuberculous lymphadenitis (TB-LAP).  This is an interesting paper but the results are very vague and that need to be addressed. They are as follows:

  • Please provide a table of all the primer sequences used in this paper. No background information on IS6110, MTP40 and alfa-antigen is provided. Why did you choose those sequences?
  • CT images are missing in this paper
  • In Table 2. What do each column represent?
  • For table 3 and 4 please show a representative figure. How did you calculate area under the curve? How did you calculate sensitivity and specificity?
  • Please elaborate further on the outcome of KW analysis and Dunn’s comparison test outcome in even though the data is insignificant.

What I liked about this paper that the authors presented the data even it is insignificant. The overall manuscript is unclear due to lack of information. The abstract does not aptly represent the manuscript. It is not as cautious and uses “should be useful biomarkers” instead of “could be useful biomarker”.  Please fix the language and maintain more cautious tone in the abstract.

I look forward to read the resubmission.

Author Response

We truly appreciate all the constructive comments and invaluable suggestions from the editor and the reviewers.

We tried to respond point by point to the remarks addressed by the reviewers.

The detailed full responses to the remarks are in the attached file.

Best regards.

Round 2

Reviewer 2 Report

The revised manuscript indeed reads much better. The authors' have satisfactorily addressed all the criticisms raised on the previous review. This paper is ready to be accepted. Congratulations on your effort!